# Effects of Swedish Massage at Different Times of the Day on Dynamic and Static Balance in Taekwondo Athletes

**DOI:** 10.3390/healthcare12020165

**Published:** 2024-01-10

**Authors:** Serdar Bayrakdaroğlu, Özgür Eken, Ramazan Bayer, Fatma Hilal Yagin, Tuba Kızılet, Recep Fatih Kayhan, Luca Paolo Ardigò

**Affiliations:** 1Department of Coaching Education, Movement and Training Sciences, School of Education and Sport, Gumushane University, Gumushane 29000, Turkey; sbayrakdaroglu@gumushane.edu.tr; 2Department of Physical Education and Sport Teaching, Faculty of Sports Sciences, Inonu University, Malatya 44000, Turkey; 3Department of Physical Education and Sport Teaching, Malatya Turgut Ozal University, Malatya 44000, Turkey; ramazan.bayer@ozal.edu.tr; 4Department of Biostatistics and Medical Informatics, Faculty of Medicine, Inonu University, Malatya 44280, Turkey; 5Department of Coaching Education, Faculty of Sports Sciences, Marmara University, Istanbul 34815, Turkey; tuba.kizilet@marmara.edu.tr; 6Department of Coaching Education, Movement and Training Sciences, Faculty of Sports Sciences, Marmara University, Istanbul 34815, Turkey; fatih.kayhan@marmara.edu.tr; 7Department of Teacher Education, NLA University College, 0166 Oslo, Norway; luca.ardigo@nla.no

**Keywords:** static balance, dynamic balance, Swedish massage, taekwondo, time of day

## Abstract

The purpose of this study is to investigate the impact of different durations of Swedish massage on the static and dynamic balance at different times of the day in taekwondo athletes. Twelve taekwondo athletes who had been practicing on a regular basis for more than 5 years participated in this study. Taekwondo athletes completed static and dynamic balance tests either after a no-massage protocol (NMP), a five-minute massage protocol (5MMP), a ten-minute massage protocol (10MMP), or a fifteen-minute massage protocol (15MMP) two times a day in the morning (08:00–12:00) and in the evening (16:00–20:00), on non-consecutive days. The findings of this study suggest that the duration of the massage has a discernible impact on dynamic balance, particularly with regard to the right foot. Taekwondo athletes who received a 10MMP or 15MMP displayed significantly improved dynamic balance compared to those in the NMP. Importantly, these improvements were independent of the time of day when the massages were administered. It underscores the potential benefits of incorporating short-duration Swedish massages into taekwondo athletes’ pre-competition routines to enhance dynamic balance. These findings highlight the potential benefits of incorporating short-duration Swedish massages into taekwondo athletes’ pre-competition routines to enhance dynamic balance, a critical component of their performance, regardless of the time of day.

## 1. Introduction

The physical aspects of taekwondo, including flexibility, strength, speed, and endurance, require a significant balance skill to be easily applied in a competitive environment. This is because competition necessitates defending against attacks from all directions by using both sides of the body [1,2]. Studies have reported that balance ability plays a significant role in maintaining posture and executing skills [3,4]. For example, Nien et al. [5] reported that the duration of kicking movements and peak velocity ranged from 0.22 to 0.31 s and 13.43 to 16.48 m/s, respectively, during competition. With these rapid and powerful kicks, maintaining balance and proper footwork are essential to avoid self-injury [1]. Additionally, Sadowski [6] reported that balance is a dominant “coordination motor skill” in elite-level martial athletes.

In competitive sports, differences in daily variations in maximum performance are of great importance [7]. Within this daily rhythm, various factors, including exposure to hot and humid environmental conditions, can influence factors that may affect sports performance [8,9]. For instance, nutrition, sleep, individual chronotype differences, rest, and body temperature at different times of the day have been shown to contribute to daily variation [10]. Many variables affecting sports performance have been reported to be influenced by daily variations, with different outcomes observed with daily variation [11,12].

However, despite the recent increase in studies on the circadian rhythm’s effects on performance development, there have been limited studies on its effects on balance performance. Gribble and Hertel [13] claimed that morning dynamic balance performance was superior to afternoon and evening performance, although Zouabi et al. [14] observed no effect of the time of day on postural static and dynamic balance. Cagno et al. [15] suggested that the static balance performance of rhythmic gymnasts remained consistent throughout the day, but dynamic balance performance peaked in the morning hours.

Beyond training schedules, understanding the effects of factors throughout the day, especially the preparation processes before competition, on athletic performance takes the interesting findings from studies on training timing with taekwondo athletes a step further. In this context, in addition to the timing of training, focusing on the contributions of preparation strategies such as warm-up routines to overall performance optimization can provide a significant contribution to sports science and practice. A review of the literature has investigated various preparation strategies before competition and has generally shown that warm-up routines are beneficial for performance [16,17]. An ideal warm-up aims to increase muscle temperature, allowing for several internal changes such as increased blood flow and optimized metabolic responses [16]. These warm-up strategies, which allow athletes to maximize their performance while minimizing fatigue, have supported the suitability of active warming in a wide range of individual sports [18,19]. However, despite active warming being the most commonly used strategy over the years, passive warming strategies have also attracted attention [16,17].

Massage, one of the passive warming methods preferred by athletes and coaches, can be applied to assist in acute preparation for performance [20]. It possesses relaxing and stimulating properties that can potentially affect an athlete’s performance in various situations, such as before, during, and after training or competition [21]. Generally defined as the manipulation of soft tissue [22], massage can be applied for the purpose of recovery, injury prevention [23], and as a pre-performance passive warming method [24].

The most commonly used type of massage in training and competitions is known as Swedish massage or Western massage. Swedish massage involves systematic manipulation of the body’s soft tissues using strokes, kneading, rubbing, and vibration during the massage [25]. These movements promote blood and lymph circulation, muscle relaxation, pain alleviation, the restoration of metabolic balance, and various other physical and mental benefits [26]. In a study by Kaplan et al. [27], massage was suggested as a way to help healthy individuals improve their balance.

Limited information exists regarding the effects of massage on balance. Sefton et al. [28] stated that massage improves both static and dynamic balance in adults. Another study also indicated the positive effects of massage on balance [29]. The limited number of studies on the acute balance performance of Swedish massage used as a passive warming method prompted the consideration of this study.

A warm-up strategy can be a good option to improve the daily performance levels of taekwondo athletes in events like taekwondo competitions, where competitions are scheduled at different times of the day and athletes participate in multiple competitions throughout the day. Current literature reviews have observed that the effects of commonly preferred massage applications by athletes on performance enhancement vary. The literature review has not found any research examining the contribution of massages applied at different times of the day and for different durations to balance development. Therefore, this study has two distinct objectives: (1) to determine the effects of Swedish massage applied to taekwondo athletes from morning to evening on balance performance, and (2) to determine the time-dependent effect of Swedish massage applied at different time intervals. Accordingly, the hypotheses of the study are defined as follows: (1) Swedish massage applied from morning to evening improves the balance performance of taekwondo athletes; (2) the effect of Swedish massage applied for different durations varies over time.

## 2. Materials and Methods

### 2.1. Participants

The power analysis indicated a minimum of twelve participants in the study. Due to the risk of participants leaving, 17 taekwondo athletes were chosen in all. Because they did not complete all of the required sessions, five of the taekwondo athletes (*n* = 5) were disqualified from the data analysis. Exclusion criteria from the study include a dermatological reaction to the oil used during massage application and an unrelated injury. In this study, 12 taekwondo athletes participated. Twelve taekwondo athletes (age: 20.660 ± 1.150 years; height: 177.750 ± 7.030 cm; body weight: 72.000 ± 15.110 kg; body fat percent: 13.500 ± 5.240; body mass index: 22.620 ± 4.440 kg/m^2^), who had been regularly exercising and participating in national competitions for a minimum of five years, voluntarily participated in the study. The inclusion criteria for the study included not having a history of physical therapy, lower extremity injuries, or surgery (operation) in the last three months and not using any anti-inflammatory medication. Measurements were taken during the taekwondo athletes’ competition season. Prior to the study, participants were provided with detailed information about the research’s content, purpose, and methodology, and voluntary consent forms were signed. Experimental procedures were approved by the Ethics Committee of the University of Gumushane, and the research followed the guidelines of Human Ethics in Research in accordance with the Helsinki Declaration (Approval Number: E-95674917-044-102285 and approval date: 22 June 2022). Before the tests, participants were allowed to perform their normal physical activities but were instructed to avoid strenuous exertions.

### 2.2. Study Design

Prior to commencing the study, two familiarization sessions were conducted to ensure the taekwondo athletes’ familiarity with both the designated massage location and the static and dynamic balance test procedures following massage protocols. Subsequently, participants arrived at the laboratory for the four specified test sessions [a no-massage protocol (NMP), a five-minute massage protocol (5MMP), a ten-minute massage protocol (10MMP), and a fifteen-minute massage protocol (15MMP)] with a minimum of 72 h between sessions. In the first session, participants’ anthropometric measurements (height, body weight, body fat percentage) were taken, and static and dynamic balance measurements were obtained as NMP, followed by static and dynamic balance measurements. In the second session, participants received a 5MMP, followed by static and dynamic balance measurements. In the third session, after a 10MMP, static and dynamic measurements were taken. Finally, in the fourth session, participants received 15MMP, followed by static and dynamic balance measurements. There was a 1 min resting period between static and dynamic balance measurements. These interventions were applied twice a day, not on consecutive days, in the morning from 08:00 to 12:00 and in the evening from 16:00 to 20:00. Participants were instructed not to eat or consume carbonated beverages for at least 2 h before the measurements. The experimental design flowchart is presented in Figure 1.

### 2.3. Massage Protocols

To ensure the hygiene of the massage table before each massage application, it was cleaned using a disinfectant, and a disposable cover was used. The massage room was adjusted to an appropriate temperature (22–26 °C). Approximately 15 mL of classic baby oil was used for each participant during the Swedish massage. Swedish massage was performed by using effleurage, friction, petrissage, and pressure techniques, with the massage strokes directed towards the heart and muscle fibers [30,31]. The Swedish massage administered to the participants was conducted by three expert masseurs, and to avoid affecting measurement outcomes, the masseurs administered massages to the same participants in each session. To ensure that the effects of the Swedish massage administered to the participants did not dissipate, the participants underwent balance tests shortly after the massage. Each device (InBody, Togu dynamic balance, and Desmotec static balance) had a dedicated expert practitioner.

### 2.4. Anthropometric Measurements

The participants’ height measurements were obtained with an electronic measuring device, Seca-769 (Seca Corporation, Hamburg, Germany), with an accuracy of 0.001 m, while their body weights and body fat percentages were measured using an electronic measurement device, Inbody 720 (Bioimpedance Body Composition Analyzer, Biospace, Seoul, Republic of Korea), with an accuracy of 0.01 kg. Body mass index (BMI) was calculated as mass (kg) divided by height (m) squared.

### 2.5. Assessment of Static Balance

The participants’ static balance values were obtained using an electronic assessment platform connected to a computer via portable and specialized software (Desmotec E-Board 1.2.0.1 version, Biella, Italy). Prior to the test, participants’ age, height, and body weight information were entered into the test application tablet, and then participants were positioned on the E-Board with bare feet and instructed to point their heels at a 30-degree angle. Participants were asked to maintain this position, aligning themselves with a plus (+) sign displayed on the test screen, and to hold this position for 50 s. The results were recorded as area (sway in the plane), avgfl (average front left balance sway), avgfr (average front right balance sway), avgfb (average front and back balance sway), and avgff (average front and front balance sway) [32].

### 2.6. Assessment of Dynamic Balance

The participants’ dynamic balance values were obtained using a portable dynamic balance and training system (Challenge Disc 2.0, TCD006, Togu, Bayern, Germany). After the participants stepped onto the test disc, a test was opened on the tablet screen using the Google Play application downloaded to the tablet. In the test procedure, participants were initially instructed to balance the disc with both feet for one minute. Subsequently, as the second test protocol, they were asked to balance the disc for 30 s with both their right and left feet [32].

### 2.7. Statistical Analysis

The study group was determined using the repeated two-way ANOVA test in the power analysis program G*Power (version 3.1.9.3, Kiel, Germany). As a result of the power analysis (power (1 − beta) = 0.80, alpha = 0.05, and partial eta square = 0.35), it was determined that at least 12 taekwondo athletes should be included in the study. Data were summarized with mean-standard deviation and median-interquartile range (IQR). Since the multivariate analysis assumptions for the groups could not be provided (multivariate normal distribution and homogeneity of variance assumptions), a two-way PERMANOVA (permutational analysis of variance) analysis with Euclidean distance as a similarity matrix was used to examine the difference within groups, the difference between groups, and the interaction effect (permutation *n* = 9999). Group differences were analyzed in terms of four groups (NMP, 5MMP, 10MMP, and 15MMP) and two measurements (morning and evening). These analyses were performed for right-foot dynamic balance, left-foot dynamic balance, dynamic balance biped and area (static balance), static balance (avgfl), static balance (avgfr), static balance (avgfb), and static balance (avgff) measurements. The Bonferroni test was used for post-hoc analyses. In addition, F statistics, significant level, and partial omega squared were assessed. Partial omega-squared values were used to show the magnitude effect size (trivial < 0.2; small ≥ 0.2; medium ≥ 0.5; large ≥ 0.8 and above). Analyses were performed using Python 3.9 and IBM SPSS Statistics for Windows version 26.0 (New York, NY, USA).

## 3. Results

Table 1 shows the changes in the dynamic balance right, left, and double foot parameters of the participants. There was no statistically significant difference between the time of day (morning and evening) in terms of right-foot dynamic balance (F_(1,11)_ =2.030, *p*_1_= 0.151). In addition, there was a statistically significant difference between the groups in terms of right-foot dynamic balance (F_(3,33)_ = 4.650, *p*_2_ = 0.004). Post-hoc analyses showed that there was a statistical difference between the NMP and 5MMP, NMP and 10MMP, and NMP and 15MMP groups in terms of right-foot dynamic balance (*p* ≤ 0.05). However, the interaction effect (time × group) for right-foot dynamic balance measurements was not statistically significant (F_(3,33)_ = 0.088, *p* = 0.964).

According to the data of the study, there was no statistically significant difference between the times (morning and evening) in terms of the left-foot dynamic balance (F_(1,11)_ = 1.751, *p*_1_ = 0.182). A statistical difference was found between the groups in terms of left-foot dynamic balance (F_(3,33)_ = 4.463, *p*_2_ = 0.006). Post-hoc analyses showed that there was a statistical difference between the NMP and 5MMP, and the NMP and 10MMP in terms of right-foot dynamic balance (*p* ≤ 0.05). The interaction effect (time × group) for left-foot dynamic balance measurement was not statistically significant (F_(3,33)_ = 0.015, *p* = 0.991).

The morning NMP protocol exhibited the highest median score of 4.371 (IQR 0.648), suggesting potentially superior balance in the morning compared to the evening NMP protocol (4.171, IQR 0.902). Significant differences were evident between groups (F_(3,33)_ = 15.932, *p* < 0.001), indicating a noteworthy impact of time-of-day variations on dynamic balance. Within the no-massage protocol subjected to different massage durations (5, 10, and 15 min), a consistent trend of improved balance was observed with longer massage durations, as indicated by the significant in-group comparison tests (*p*_3_ < 0.001). As a result, dynamic balance improved significantly in the 5, 10, and 15 min massage protocols, and the level of improvement increased as the minutes of massage increased. However, the intergroup comparisons yielded mixed results, with certain pairs showing statistically significant differences and others displaying no significant variance in dynamic balance.

Table 2 shows the changes in the area (static balance), static balance (avgfl), static balance (avgfr), static balance (avgfb), and static balance (avgff) parameters of the participants. There was no statistically significant difference between the times of day (morning and evening) in terms of area (static balance) (F_(1,11)_ = 0.025, *p*_1_ = 0.870). In addition, there was no statistically significant difference between the groups in terms of area (static balance) (F_(3,33)_ = 1.860, *p*_2_ = 0.130). The interaction effect (time × group) for area (static balance) measurements was not statistically significant (F_(3,33)_ = 0.138, *p* = 0.930).

According to the data of the study, there was no statistically significant difference in terms of static balance (avgfl) between the times of day (morning and evening) (F_(1,11)_ = 0.017, *p*_1_ = 0.880). In addition, there was no statistically significant difference between the groups in terms of static balance (avgfl) (F_(3,33)_ = 0.026, *p*_2_ = 0.990). The interaction effect (time × group) for static balance (avgfl) measurements was not statistically significant (F_(3,33)_ = 0.020, *p* = 0.990).

In addition, there was no statistically significant difference between the times of day (morning and evening) in terms of static balance (avgfr) (F_(1,11)_ = 0.130, *p*_1_ = 0.710). In addition, there was no statistically significant difference between the groups in terms of static balance (avgfr) (F_(3,33)_ = 0.043, *p*_2_ = 0.982). The interaction effect (time × group) for static balance (avgfr) measurements was not statistically significant (F_(3,33)_ = 0.024, *p* = 0.990). There was no statistically significant difference in terms of static balance (avgfb) between the times of day (morning and evening) (F_(1,11)_ = 0.211, *p*_1_ = 0.630). In addition, there was no statistically significant difference between the groups in terms of static balance (avgfb) (F_(3,33)_ = 0.063, *p*_2_ = 0.982). The interaction effect (time × group) for static balance (avgfb) measurements was not statistically significant (F_(3,33)_ = 0.064, *p* = 0.973). Time of day (morning and evening), groups, and interaction effects were not significant for static balance (avgff).

## 4. Discussion

The purpose of this study is to investigate the impact of different durations of Swedish massage on the static and dynamic balance of taekwondo athletes, with a specific focus on how these effects vary at different times of the day. The study aims to provide valuable insights into the potential benefits of incorporating short-duration Swedish massages into taekwondo athletes’ pre-competition routines as a means to enhance their dynamic balance, a crucial element of their overall performance.

The findings of this study provide compelling evidence of the influence of massage duration on dynamic balance in taekwondo athletes. Specifically, it was observed that taekwondo athletes who received either a 10 min or a 15 min Swedish massage displayed significantly improved dynamic balance compared to those who did not receive any massage protocols (NMP). This observation suggests that the duration of the massage plays a discernible role in enhancing dynamic balance, particularly in relation to the right foot.

The positive effects of a 10 min or 15 min massage on dynamic balance can be attributed to various factors associated with Swedish massage techniques. These techniques, which involve a combination of effleurage, petrissage, friction, tapotement, and kneading, are known to stimulate blood flow, relax muscles, and reduce muscle tension. As a result, they may contribute to improved proprioception and neuromuscular control, ultimately leading to enhanced dynamic balance [29].

An important finding in this study is that the improvements in dynamic balance observed in taekwondo athletes who received 10 min or 15 min massages were independent of the time of day when the massages were administered. This suggests that the benefits of Swedish massage on dynamic balance can be realized both in the morning (08:00–12:00) and in the evening (16:00–20:00). This finding is significant as it implies that taekwondo athletes can integrate massage into their pre-competition routines regardless of the competition schedule, ensuring consistent improvements in dynamic balance.

In our study, it was observed that the 5 min massage had significantly higher right-foot dynamic balance measurements compared to the 10 min and 15 min massage groups. Additionally, the 5 min massage showed significantly higher left-foot dynamic balance measurements compared to both the 5 min and 10 min massage groups. Moreover, the 5 min massage exhibited significantly higher dynamic balance measurements for both feet compared to the 10 min and 15 min massage groups. Upon reviewing the literature, it was determined that there is a lack of research focusing on the duration aspect of diurnal variation massages, specifically regarding their effects on balance. Furthermore, studies examining the impact of different massage techniques at varying durations on balance and functional capacity are limited in number. Regarding the overall assessment of massage durations, Park et al. [33] investigated the effects of calf massage on balance and applied a 10 min massage protocol, reporting significant increases without statistical significance. These findings align with the results of our study, where a 10 min massage protocol was applied, yielding similar outcomes. Yu et al. [34] tested the impact of a 40 min manual massage technique on balance and found no significant improvement in balance immediately after massage. However, the absolute results demonstrated a tendency towards clinical improvement. Sefton et al. [35] examined the effects of a combination session of different therapeutic massage techniques (58 min in total) on stability and observed an improvement in balance in both conditions (bipedal and unipedal). Zhang et al. [36] observed a statistically significant increase in dynamic balance performance after 6 min massage sessions in recreationally active adults. Conversely, Junker and Stöggl [37] reported that 30 min massage sessions had no effect on dynamic balance performance. Similarly, Grabow et al. [38] found that massage therapy did not alter static balance performance in young adults. Shin and Sung [39] found that a 15 min massage therapy applied to the gastrocnemius muscle increased proprioception. Additionally, Naderi et al. [40] reported that a 15 min massage reduced muscle pain and balance loss in their study. Massage is one of the most commonly used recovery techniques to alleviate issues resulting from muscle damage, aiding in the removal of interstitial inflammatory mediators, edema, and muscle tension [41,42,43]. It can enhance proprioception and physical performance [42]. In light of these varying findings in the literature, it can be suggested that the effectiveness of massage may depend on factors such as the timing, duration, frequency, and type of massage applied. Some studies indicate that shorter massage sessions (5–15 min) may be more effective in improving specific performance parameters, while longer sessions (15 min or more) may not yield significant improvements [40,44,45,46].

Our study aimed to compare groups in terms of diurnal variation, with a hypothesis focusing on this aspect. It was determined that there were no significant differences in either static or dynamic balance levels concerning diurnal variation. Upon conducting a literature review, it was evident that our study is the first to compare the effects of massage on balance performance in taekwondo athletes throughout the day, assessing diurnal variation. In contrast to our study, a study investigating the impact of diurnal variation on anaerobic performance found significant effects in different samples [47]. Reilly and Garrett [48] found no significant differences between morning and evening performance measurements in their study on the influence of diurnal variation. Another study by Rommel et al. [49] reported significant results in diurnal variation applications. Pullinger et al. [50] noted a significant effect of diurnal variation on repeated sprint performance. Other studies highlight the influence of environmental factors on the effect of diurnal variation on performance [51]. Some authors also explain that diurnal variation is a significant endogenous component in muscle force production [52,53]. In most studies, irrespective of the effect of massage, it is clear that performance varies depending on the time of day [54,55]. These studies have consistently observed that maximum performance is achieved in the afternoon [54,56,57]. Accordingly, our study’s results align with previous findings but also present differences. Variations in the composition of study groups and the performance measures assessed in our study compared to the examples provided may account for these differing results. Additionally, it is worth considering that the effectiveness of massage techniques applied in our study may have contributed to the disparities in findings when compared to existing literature.

While this study provides valuable insights into the impact of Swedish massage duration on dynamic balance in taekwondo athletes, several limitations must be acknowledged. First, the study did not explore potential individual variations in response to massage, including factors such as taekwondo athletes’ baseline balance abilities, muscle composition, or injuries. These individual differences could influence how taekwondo athletes respond to different massage durations. Second, the study focused solely on the immediate effects of massage on dynamic balance and did not investigate potential cumulative effects over an extended period. It remains unclear whether repeated short-duration massages would yield different or more pronounced results. Third, in our study, the randomization method was not used. This could be a limitation. Finally, the study did not assess the psychological aspects of the taekwondo athletes’ experiences, such as their perception of relaxation and readiness after different massage durations, which could have practical implications for coaches and athletes. Future research should consider these factors to provide a more comprehensive understanding of the relationship between Swedish massage and athletic performance in taekwondo and other sports.

## 5. Conclusions

The results of this study have practical implications for taekwondo athletes and their coaches. Enhanced dynamic balance is critical for taekwondo athletes as it directly impacts their performance, particularly during fast and dynamic movements in competition. Therefore, incorporating short-duration Swedish massages, specifically 10MMP or 15MMP protocols, into pre-competition routines can be recommended as a strategy to optimize dynamic balance. These massages can be performed in a time-efficient manner, making them feasible for athletes even on a tight schedule of competition days. Coaches and athletes can work together to create a structured pre-competition routine that includes massage as a regular component to ensure consistent improvements in dynamic balance. Moreover, the independence of massage effects from the time of day allows for flexibility in scheduling, accommodating early morning and late evening competitions. In conclusion, this study underscores the potential benefits of incorporating short-duration Swedish massages into taekwondo athletes’ pre-competition routines to enhance dynamic balance, irrespective of the time of day. These findings provide valuable insights into optimizing the performance of taekwondo athletes and can serve as a foundation for further research into the application of massage techniques in sports performance enhancement. 

## Figures and Tables

**Figure 1 healthcare-12-00165-f001:**
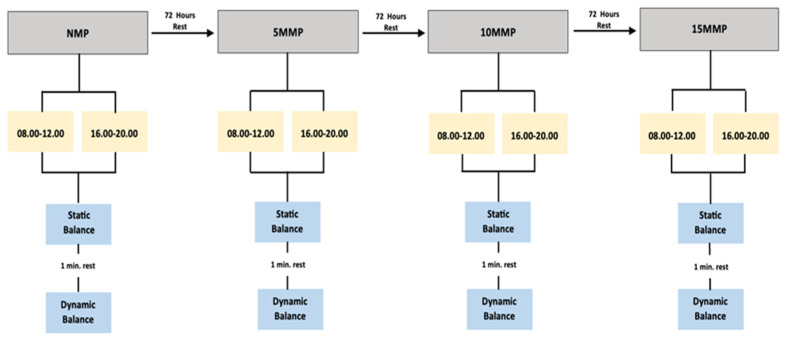
Experimental design of the research.

**Table 1 healthcare-12-00165-t001:** Comparison of the measured values of dynamic balance right, left, and double foot.

Groups	Median (IQR)	Between Measurements	Between Groups	Interaction
		F Value	F Value	
		*p*_1_ Value	*p*_2_ Value	
Right-Foot Dynamic Balance		**ES**	**ES**	
NMP—morning	3.845 (0.792)	F_(1,11)_ = 2.030	F_(3,33)_ = 4.650	F_(3,33)_ = 0.088
5MMP—morning	3.145 (0.935)	*p*_1_ = 0.151	***p***_2_ **= 0.004 *** **ES = 0.228**	*p* = 0.964
10MMP—morning	3.040 (1.098)	ES = Not effect	NMP-5MMP (***p***_3_ **< 0.001 ***)	ES = Not effect
15MMP—morning	3.190 (0.757)		NMP-10MMP (***p***_3_ **= 0.004 ***)	
NMP—evening	3.545 (0.89)		NMP-15MMP (***p***_3_ **= 0.011 ***)	
5MMP—evening	2.665 (0.678)		5MMP-10MMP (*p*_3_ = 0.752)	
10MMP—evening	3.040 (0.628)		5MMP-15MMP (*p*_3_ = 0.383)	
15MMP—evening	2.845 (0.862)		10MMP-15MMP (*p*_3_ = 0.631)	
Left-Foot Dynamic Balance				
NMP—morning	3.811 (1.075)	F_(1,11)_ = 1.751	F_(3,33)_ = 4.463	F_(3,33)_ = 0.015
5MMP—morning	2.745 (1.331)	*p*_1_ = 0.182	***p***_2_ **= 0.006 *** **ES = 0.219**	*p* = 0.991 ES = Not effect
10MMP—morning	2.863 (1.052)	ES = Not effect	NMP-5MMP (***p***_3_ **= 0.001 ***)	
15MMP—morning	3.055 (1.152)		NMP-10MMP (***p***_3_ **= 0.002 ***)	
NMP—evening	3.672 (1.152)		NMP-15MMP (*p*_3_ = 0.131)	
5MMP—evening	2.395 (0.910)		5MMP-10MMP (*p*_3_ = 0.642)	
10MMP—evening	2.680 (0.870)		5MMP-15MMP (*p*_3_ = 0.102)	
15MMP—evening	2.952 (1.645)		10MMP-15MMP (*p*_3_ = 0.203)	
Double-Foot Dynamic Balance				
NMP—morning	4.371 (0.648)	F_(1,11)_ = 0.111	F_(3,33)_ = 15.932	F_(3,33)_ = 0.274 *p* = 0.844 ES = Not effect
5MMP—morning	2.725 (0.998)	*p*_1_ = 0.743 ES = Not effect	** *p* ** _2_ ** < 0.001** **ES = 0.547**	
10MMP—morning	3.055 (1.243)		NMP-5MMP (***p***_3_ **< 0.001 ***)	
15MMP—morning	3.132 (1.111)		NMP-10MMP (***p***_3_ **< 0.001 ***)	
NMP—evening	4.171 (0.902)		NMP-15MMP (***p***_3_ **< 0.001 ***)	
5MMP—evening	2.825 (0.912)		5MMP-10MMP (*p*_3_ = 0.790)	
10MMP—evening	2.845 (1.735)		5MMP-15MMP (*p*_3_ = 0.082)	
15MMP—evening	3.365 (0.947)		10MMP-15MMP (*p*_3_ = 0.184)	

IQR: Interquartile range, *p*_1_ value; significance test result between measurements, *p*_2_ value; intergroup PERMANOVA significance test result, *p*_3_ value; the results of the in-group comparison significance test; ES: effect size; Note: Values in bold and * indicate statistically significant results.

**Table 2 healthcare-12-00165-t002:** Comparison of the measured values of area (static balance), static balance (avgfl), static balance (avgfr), static balance (avgfb), and static balance (avgff).

Groups	Median (IQR)	Between Measurements	Between Groups	Interaction
		F Value	F Value	
		*p*_1_ Value	*p*_2_ Value	
Area (Static Balance)		ES	ES	
NMP—morning	19.000 (25.250)	F_(1,11)_ = 0.025	F_(3,33)_ = 1.860	F_(3,33)_ = 0.138
5MMP—morning	18.000 (20.251)	*p*_1_ = 0.870 ES = Not effect	*p*_2_ = 0.130 ES = Not effect	*p* = 0.930 ES = Not effect
10MMP—morning	27.000 (12.500)			
15MMP—morning	39.000 (37.012)			
NMP—evening	22.000 (11.250)			
5MMP—evening	19.000 (20.250)			
10MMP—evening	21.000 (19.000)			
15MMP—evening	28.500 (34.000)			
Static Balance (avgfl)				
NMP—morning	36.450 (7.400)	F_(1,11)_ = 0.017	F_(3,33)_ = 0.026	F_(3,33)_ = 0.020
5MMP—morning	34.600 (9.050)	*p*_1_ = 0.880 ES = Not effect	*p*_2_ = 0.990 ES = Not effect	*p* = 0.990 ES = Not effect
10MMP—morning	34.850 (8.550)			
15MMP—morning	34.950 (7.800)			
NMP—evening	34.800 (8.350)			
5MMP—evening	35.050 (8.275)			
10MMP—evening	35.000 (9.225)			
15MMP—evening	35.050 (9.550)			
Static Balance (avgfr)				
NMP—morning	36.150 (7.100)	F_(1,11)_ = 0.130	F_(3,33)_ = 0.043	F_(3,33)_ = 0.024
5MMP—morning	34.300 (9.600)	*p*_1_ = 0.711 ES = Not effect	*p*_2_ = 0.982 ES = Not effect	*p* = 0.990 ES = Not effect
10MMP—morning	34.750 (8.300)			
15MMP—morning	34.950 (8.875)			
NMP—evening	34.300 (8.725)			
5MMP—evening	34.000 (8.125)			
10MMP—evening	34.400(9.050)			
15MMP—evening	34.800 (9.025)			
Static Balance (avgfb)				
NMP—morning	37.100 (7.700)	F_(1,11)_ = 0.211	F_(3,33)_ = 0.063	F_(3,33)_ = 0.064
5MMP—morning	34.450 (9.825)	*p*_1_ = 0.630 ES = Not effect	*p*_2_ = 0.982 ES = Not effect	*p* = 0.973 ES = Not effect
10MMP—morning	35.000 (8.700)			
15MMP—morning	35.200 (9.925)			
NMP—evening	34.400 (8.875)			
5MMP—evening	34.850 (7.300)			
10MMP—evening	35.150 (8.850)			
15MMP—evening	35.200 (8.375)			
Static Balance (avgff)				
NMP—morning	35.300 (7.025)	F_(1,11)_ = 0.151	F_(3,33)_ = 0.026	F_(3,33)_ = 0.007
5MMP—morning	34.700 (9.100)	*p*_1_ = 0.690 ES = Not effect	*p*_2_ = 0.990 ES = Not effect	*p* = 0.991 ES = Not effect
10MMP—morning	34.600 (8.675)			
15MMP—morning	34.750 (9.075)			
NMP—evening	34.800 (8.750)			
5MMP—evening	34.500 (8.200)			
10MMP—evening	34.700 (8.425)			
15MMP—evening	33.700 (7.650)			

Area: sway in the plane; avgfl: average front left balance sway; avgfr: average front right balance sway; avgfb: average front and back balance sway; avgff: average front and front balance sway; IQR: interquartile range, *p*_1_ value; significance test result between measurements, *p*_2_ value; intergroup PERMANOVA significance test result; ES: effect size.

## Data Availability

The datasets generated and/or analyzed during the current research are available from the corresponding author on reasonable request.

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
