# Peer review of "Effects of Swedish Massage at Different Times of the Day on Dynamic and Static Balance in Taekwondo Athletes"

_healthcare, 2024, doi:10.3390/healthcare12020165_

Round 1

Reviewer 1 Report

Comments and Suggestions for Authors

Abstract

1.        The discrepancy between the time duration of massage might bring up a bias of placebo.

2.        Please provide explicit information regarding the dynamic and static balance tests

3.        Please name the Taekwondo athletes or Athletes consistently throughout the text.

4.        Statistical output to significant levels should be reported.

Introduction

1.        The introduction needs to be revised. The authors need to emphase the relationship between the balance control in Taekwondo performance and diurnal effect of balance capacity. There is no need to address how the anaerobic performance is critical in this sport event (i.e. line 40-49, line 66-103).

2.        Statements between line 104-113 is not related to the research aims. Suggested to remove.

3.        Review the current knowledge in Judo is not associated the sample pool of the study.

4.        The rationale to use Swedish massage as an intervention is unclear in this session. How the implantation is optimal for balance performance in different time of the day?

Materials and methods

1.        A major question/limitation for the experimental trials was not processed in a randomized order.

2.        The participants completed the day-time trial between 08:00-1200 and night-time trial between 16:00-20:00. How the research group ensure the time interval between day-time measure and night-time measure consistently engaged among the testing days? Time intervals may vary from individual and individual as well.

3.        Line 158-9, it is questionable that partial eta square was set at 0.35 rather than 0.40.

4.        Line 162-3, personal issues for the exclusion of 5 participants should be reported here.

5.        Line 179, any specific reason for 2 familiarization sessions instead of one.

Results

1.        Suggested to add the effect size of comparison.

2.        How the author present median and interquartile range for repeated measure of ANOVA?

Conclusion

1.        What do you mean “busy competition days”, please revise as “tight schedule of competition days” or all the other terms in sport training.

Tables

1.        Interquartile range should be presented as “25%-75% of range”.

2.        Decimal numbers should be reported consistently. Perhaps include hundreds of decimal parts.

3.        Consider to converge the Table 1-Table 3 and Table 4-Table 8.

The reference list is massive, I strongly suggest reducing the number of references for this study.

Comments on the Quality of English Language

Suggested to improve the English presentation.

Author Response

  1. Review Comments and Suggestions for Authors

Abstract

  1. The discrepancy between the time duration of massage might bring up a bias of placebo.

Answer: Dear Referee, thank you for your critique. We are investigating the effects of massage durations on dynamic balance in our study, considering the possibility of a placebo effect. To minimize the placebo effect, we ensured that participants were unaware of the durations of the massages, aiming to diminish the impact of expectations related to the duration of the massage. Furthermore, supporting the observed improvements in dynamic balance with statistical analyses as the duration of the massage increases indicates that these improvements are based on physiological foundations rather than the placebo effect. However, we acknowledge the importance of including placebo-controlled groups in future studies to further examine the placebo effect in more detail.

  1. Please provide explicit information regarding the dynamic and static balance tests

Answer: Detailed information about dynamic and static balance tests has been thoroughly explained in the introduction and methodology sections. However, due to the journal's guidelines limiting the abstract section to a maximum of 200 words, this part hasn't been provided in detail there.

  1. Please name the Taekwondo athletes or Athletes consistently throughout the text.

Answer: It was changed to Taekwondo athletes

  • Statistical output to significant levels should be reported.

Answer: Due to the maximum word limit of 200 words for the abstract section in the journal's guidelines, detailed findings have been provided in the Results section

Introduction

  1. The introduction needs to be revised. The authors need to emphase the relationship between the balance control in Taekwondo performance and diurnal effect of balance capacity. There is no need to address how the anaerobic performance is critical in this sport event (i.e. line 40-49, line 66-103).

Answer: The introduction has been revised

  1. Statements between line 104-113 is not related to the research aims. Suggested to remove.

Answer: Dear referee, our study investigates the warm up effect (passive warm up) of Swedish massage applied at different durations and times of the day on static and dynamic balance performance. Therefore, our warm-up protocol is directly related to the aim of our study. Additionally, the introduction section has been revised

  1. Review the current knowledge in Judo is not associated the sample pool of the study.

Answer: The sections related to Judo have been removed

  1. The rationale to use Swedish massage as an intervention is unclear in this session. How the implantation is optimal for balance performance in different time of the day?

Answer: Revised, thank you

Materials and methods

  1. A major question/limitation for the experimental trials was not processed in a randomized order.

Answer: You're right, the randomization method wasn't used in the study. This was added to the limitations section.

  1. The participants completed the day-time trial between 08:00-1200 and night-time trial between 16:00-20:00. How the research group ensure the time interval between day-time measure and night-time measure consistently engaged among the testing days? Time intervals may vary from individual and individual as well.

Answer: Dear Reviewer, we understand your concerns regarding the consistency of timing and would like to assure you that specific measures were taken in our study to address this. We standardized the time intervals for each participant; morning sessions were scheduled between 08:00-12:00 and evening sessions between 16:00-20:00. On each testing day, we ensured a minimum gap of eight hours between these sessions. This interval was critical to allow participants sufficient rest between the two trials, thereby ensuring the reliability of the experiment. Additionally, participants had at least one day off between trials, which helped maintain a consistent interval between each session. All these measures were carefully implemented to minimize individual variations in time intervals and to preserve the integrity of the experiment.

  1. Line 158-9, it is questionable that partial eta square was set at 0.35 rather than 0.40.

        Answer: It was calculated from the smallest effect size; therefore, it was taken as 0.35

  • Line 162-3, personal issues for the exclusion of 5 participants should be reported here.

Answer: Exclusion criteria were added to the study.

Line 179, any specific reason for 2 familiarization sessions instead of one.

Answer: Participants were given the opportunity to try the devices twice in order to gain more experience with them

Results

  1. Suggested to add the effect size of comparison.

Answer:: Thank you so much for your valuable suggestion. Upon your suggestion, effect sizes were calculated with partial omega squared, which is suitable for PERMANOVA, and effect sizes of comparisons that were significant and had an effect were reported.

  1. How the author present the median and interquartile range for repeated measures of ANOVA?

Answer:: Two-way PERMANOVA, a non-parametric test, was used in the study and was stated in the statistical analyses. In analyses using non-parametric tests, it is more accurate to use median and IQR in descriptive statistics.

Conclusion

  1. What do you mean “busy competition days”, please revise as “tight schedule of competition days” or all the other terms in sport training.

Answer: Revised as ‘’tight schedule of competition days’’.

Tables

  1. Interquartile range should be presented as “25%-75% of range”.

Answer: The Interquartile Range (IQR) is a statistical measure that represents the range of the middle 50% of a dataset. To calculate the IQR, you first need to order the dataset from smallest to largest and then find the values of the first quartile (Q1) and the third quartile (Q3). The IQR is the difference between Q3 and Q1.

Mathematically, the formula for the interquartile range is:

IQR=Q3−Q1

Q1 (First Quartile): Represents the value below which 25% of the data falls. It is the median of the lower half of the dataset.

Q3 (Third Quartile): Represents the value below which 75% of the data falls. It is the median of the upper half of the dataset.

The IQR is useful in describing the spread or dispersion of a dataset, especially in the presence of outliers. It provides a measure of variability that is not affected by extreme values or outliers in the data. The IQR is often used in conjunction with box plots, where the "box" represents the interquartile range, and whiskers extend to the minimum and maximum values within a specified range.

For this reason, IQR is stated this way in the study. And this is the correct form according to IQR calculation.

  1. Decimal numbers should be reported consistently. Perhaps include hundreds of decimal parts.

Answer: Revised.

  1. Consider to converge the Table 1-Table 3 and Table 4-Table 8.

Answer: In the Results section, upon your suggestion, the tables were combined and the results were arranged accordingly.

  1. The reference list is massive, I strongly suggest reducing the number of references for this study.

Answer: Revised.

Reviewer 2 Report

Comments and Suggestions for Authors

The article is interesting, with an unconventional topic, with an emphasis on the application of the results in sports practice, but I would recommend improving a few elements that could improve its quality.

1. The introduction should be shortened and more closely related to the topic of the work, avoiding scattering the main idea into threads poorly related to the goal and hypotheses.

2. It would be better to present the study group by presenting the data in a table rather than in the text.

3. The discussion should be rebuilt, taking into account the exclusion of repetitions of content from the introduction, material and methods, and results sections. It would also be necessary to eliminate lists of research and arguments that do not correspond well with the purpose of the work and give the impression of multiplying the text.

4. Conclusions should be specific, more closely related to the research results and free from unnecessary content about the benefits of massage.

Author Response

  1. Review Comments and Suggestions for Authors

The article is interesting, with an unconventional topic, with an emphasis on the application of the results in sports practice, but I would recommend improving a few elements that could improve its quality.

  1. The introduction should be shortened and more closely related to the topic of the work, avoiding scattering the main idea into threads poorly related to the goal and hypotheses.

Answer: Revised.

  1. It would be better to present the study group by presenting the data in a table rather than in the text.

Answer: Revised.

  1. The discussion should be rebuilt, taking into account the exclusion of repetitions of content from the introduction, material and methods, and results sections. It would also be necessary to eliminate lists of research and arguments that do not correspond well with the purpose of the work and give the impression of multiplying the text.

Answer: Revised.

  1. Conclusions should be specific, more closely related to the research results and free from unnecessary content about the benefits of massage.

Answer: Revised.

Reviewer 3 Report

Comments and Suggestions for Authors

I would like to thank the authors for allowing me to review this study. This study shows promising results that should be furtherly developed on studies conducted on a bigger scale. I do however have a few suggestions.

The objective of the article is unclear, could you please rephrase de title? Is the variance of balance during the day the main focus? Or is the swedish massage? Also, if possible, please include the study type at the end of the title. For example, "Effects os Swedish massage at different points of the day on dynamic and static balance in Taekwondo Athletes"

Please use the template provided by Healthcare to complete the abstract. Abstract must be divided by sections

The Introduction is way too long, authors must be more concise. It is clear that authors have made their efforts when researching about the topic, but they must be able to summarize the information.

Lines 125-138. I believe it is important to develop an explain why is massage effective on balance. This is the important part of the Introduction, the justification of why are you trying to assess massage on balance performance. Please explain the mechanisms that are triggered to make patients improve balance when being massaged, this would give a more clear rationale to the study.

Lines 157-158. This should be presented under a subsection calles "Statistical analysis"

Line 159. At least 12 participants per group? Or overall? 12 participants overall seems to be a very small sample.

Line 162-163. This is supposed to be presented under "Results" section

Didn't you consider gender or foot dominance as a relevant demographic characteristic? Are all the participants male taekwondo fighters? Are they all right-footed?

"not having a history of physical therapy" This means haven't been treated by a physical therapist ever? How do you assess that and why is it an exclusion criteria?

Not having a control group, or even different groups to assess differences between massage protocols is a limitation in this study. A different study desing might have presented more robust results

Line 246, grammar mistake "analyzes". Please correct.

Line 368-369 "they may contribute to improved proprioception and neuromuscular control, 368 ultimately leading to enhanced dynamic balance" This statement should be referenced and furtherly developed, as it looks like the explanation of your study results.

Again, thank you very much for allowing me to review your manuscript.

Author Response

  1. Review Comments and Suggestions for Authors

I would like to thank the authors for allowing me to review this study. This study shows promising results that should be furtherly developed on studies conducted on a bigger scale. I do however have a few suggestions.

The objective of the article is unclear, could you please rephrase de title? Is the variance of balance during the day the main focus? Or is the swedish massage? Also, if possible, please include the study type at the end of the title. For example, "Effects os Swedish massage at different points of the day on dynamic and static balance in Taekwondo Athletes"

Please use the template provided by Healthcare to complete the abstract. Abstract must be divided by sections

The Introduction is way too long, authors must be more concise. It is clear that authors have made their efforts when researching about the topic, but they must be able to summarize the information.

Lines 125-138. I believe it is important to develop an explain why is massage effective on balance. This is the important part of the Introduction, the justification of why are you trying to assess massage on balance performance. Please explain the mechanisms that are triggered to make patients improve balance when being massaged, this would give a more clear rationale to the study.

Answer: Dear referee, the introduction section has been revised. However, since our study also considers different times of the day as a parameter, this has been indicated in the study title

Lines 157-158. This should be presented under a subsection calles "Statistical analysis"

Answer: Information about the power analysis procedure is included under the heading of statistical analysis.

Line 159. At least 12 participants per group? Or overall? 12 participants overall seems to be a very small sample.

Answer: A total of 12 participants participated and participants followed different protocols. According to the power analysis results, the sample size was sufficient.

“The study group was determined using the repeated two-way ANOVA test in the power analysis program G*Power (version 3.1.9.3, Germany). As a result of the power analysis (power (1-beta) = 0.80, alpha=0.05, and partial eta square=0.35), it was determined that at least 12 taekwondo athletes should be included in the study.”

Line 162-163. This is supposed to be presented under "Results" section

Didn't you consider gender or foot dominance as a relevant demographic characteristic? Are all the participants male taekwondo fighters? Are they all right-footed?

Answer: Revised

"not having a history of physical therapy" This means haven't been treated by a physical therapist ever? How do you assess that and why is it an exclusion criteria?

Answer: As a response to the referee's question, "not having a history of physical therapy" indeed refers to participants who have never been treated by a physical therapist. This exclusion criterion is assessed through participants' self-reported medical histories and screening interviews. We selected this criterion to ensure a homogeneous participant group, unaffected by previous therapeutic interventions that could influence the outcomes of our study. Excluding those with a history of physical therapy aims to minimize variability and eliminate potential confounding factors, thereby providing clearer insights into the direct effects of the interventions tested in our research.

Not having a control group, or even different groups to assess differences between massage protocols is a limitation in this study. A different study desing might have presented more robust results

Answer: Dear Reviewer, thank you for your critique. We are investigating the effects of massage durations on dynamic balance in our study, considering the possibility of a placebo effect. To minimize the placebo effect, we ensured that participants were unaware of the durations of the massages, aiming to diminish the impact of expectations related to the duration of the massage. Furthermore, supporting the observed improvements in dynamic balance with statistical analyses as the duration of the massage increases indicates that these improvements are based on physiological foundations rather than the placebo effect. However, we acknowledge the importance of including placebo-controlled groups in future studies to further examine the placebo effect in more detail.

Line 246, grammar mistake "analyzes". Please correct.

Answer: Revised

Line 368-369 "they may contribute to improved proprioception and neuromuscular control, 368 ultimately leading to enhanced dynamic balance" This statement should be referenced and furtherly developed, as it looks like the explanation of your study results.

Again, thank you very much for allowing me to review your manuscript.

Reviewer 4 Report

Comments and Suggestions for Authors

Thank you very much for the work. Here is some suggestions that may improve the manuscript.

Introduction

-It may be too long. This part can be shorten. Please focus more on the balance ability to Taekwondo's performance as well as duration of massage. 

Methods

-L. 165-266 please present the number with one decimal

-This investigation took part during competition, was there any specific balance training along this measurement period? If yes, please provide the details.

-Study design can be shorten as some part presented in Fig 1 and please remove the ")" in L.181

-Massage protocol that given to the participant, whether it was applied distally toward the muscle?

Results

-It is suggested to be shorten and it would be better if Table 1-3 & Table 4-8 could be merged together. 

-Please be careful to report the same info in the Table and in the text.

Discussion

-L.359-360 Please reconsider to say only 10-15 min massage significantly improve dynamic balance as the results showed that 5 min also substantially improved..

-Please provide the relevant citation for L.368-369

-Please recheck L. 388-389 5 min to 5 min and 10 min

-It would be better to discuss more static balance which not affected by the massage application.

-Please specify the performance stated that for ref # 72,73,75.

Author Response

  1. Review Comments and Suggestions for Authors

Thank you very much for the work. Here is some suggestions that may improve the manuscript.

Introduction

-It may be too long. This part can be shorten. Please focus more on the balance ability to Taekwondo's performance as well as duration of massage. 

Answer: Revised

Methods

-L. 165-266 please present the number with one decimal

Answer: Revised

-This investigation took part during competition, was there any specific balance training along this measurement period? If yes, please provide the details.

-Study design can be shorten as some part presented in Fig 1 and please remove the ")" in L.181

Answer: Revised.

-Massage protocol that given to the participant, whether it was applied distally toward the muscle?

Answer: Yes, it was applied distally, similar to the procedure in Swedish massage.

Results

-It is suggested to be shorten and it would be better if Table 1-3 & Table 4-8 could be merged together. 

Answer: Revised.

-Please be careful to report the same info in the Table and in the text.

Answer: Revised.

Discussion

-L.359-360 Please reconsider to say only 10-15 min massage significantly improve dynamic balance as the results showed that 5 min also substantially improved..

Answer: Revised

-Please provide the relevant citation for L.368-369

Answer: Revised

-Please recheck L. 388-389 5 min to 5 min and 10 min

Answer: Revised

-It would be better to discuss more static balance which not affected by the massage application.

Answer: Revised

-Please specify the performance stated that for ref # 72,73,75.

Answer: Revised

Round 2

Reviewer 1 Report

Comments and Suggestions for Authors

The authors have done great job to revise the manuscript. However, before publication, I would like to ask the authors to add the report of degrees of freedom in the F statistic.

Comments on the Quality of English Language

 Minor editing of English language required

Author Response

REVIEWER 1 COMMENTS AND AUTHORS RESPONSE

Paper title: Effects of Swedish massage at different times of the day on dynamic and static balance in Taekwondo Athletes

Dear Editor,

The authors would like to thank the reviewers for their precious time and invaluable comments. We have carefully addressed all the comments. Our detailed, point-by-point responses to the reviewer comments are given below. Additionally, we have carefully revised the manuscript to ensure that the text is optimally phrased and free from typographical and grammatical errors. Best regards,

Author's Reply to the Review Report (Reviewer 1)

Reviewer 1 Comment: The authors have done great job to revise the manuscript. However,

before publication, I would like to ask the authors to add the report of degrees of freedom in

the F statistic.

Reviewer 1 Answer: Thanks for the valuable suggestion and added.

Reviewer 3 Report

Comments and Suggestions for Authors

Dear authors, I appreciate your efforts in addressing the majority of the comments I provided. Nevertheless, there are still certain aspects that remain unaddressed. I find it perplexing that these points were not attended to, especially considering the absence of any explanatory justification for their non-resolution.

"The objective of the article is unclear, could you please rephrase de title? Is the variance of balance during the day the main focus? Or is the swedish massage? Also, if possible, please include the study type at the end of the title. For example, "Effects os Swedish massage at different points of the day on dynamic and static balance in Taekwondo Athletes". 

I strongly believe that attention should be given to the title, as it is excessively lengthy and lacks information about the type of study. Kindly take into consideration my suggestion, or provide an explanation if you choose not to address this comment.

"Please use the template provided by Healthcare to complete the abstract. Abstract must be divided by sections"

This point holds significant importance. It is imperative to adhere to the Guide for Authors of the respective journal when submitting a manuscript. In the case of Healthcare, a template is provided for adherence. However, I observe a deviation from these guidelines in your submission, particularly in the structure of the abstract. I kindly request that you address this issue in your revision.

Thank you very much again for allowing me to review your manuscript.

Author Response

REVIEWER-3 COMMENTS AND AUTHORS RESPONSE

Paper title: Effects of Swedish massage at different times of the day on dynamic and static balance in Taekwondo Athletes

Dear Editor,

The authors would like to thank the reviewers for their precious time and invaluable comments. We have carefully addressed all the comments. Our detailed, point-by-point responses to the reviewer comments are given below. Additionally, we have carefully revised the manuscript to ensure that the text is optimally phrased and free from typographical and grammatical errors. Best regards,

Author's Reply to the Review Report (Reviewer 3)

Dear authors, I appreciate your efforts in addressing the majority of the comments I provided. Nevertheless, there are still certain aspects that remain unaddressed. I find it perplexing that these points were not attended to, especially considering the absence of any explanatory justification for their non-resolution.

Reviewer 3 Comment-1: "The objective of the article is unclear, could you please rephrase de title? Is the variance of balance during the day the main focus? Or is the swedish massage? Also, if possible, please include the study type at the end of the title. For example, "Effects os Swedish massage at different points of the day on dynamic and static balance in Taekwondo Athletes".  I strongly believe that attention should be given to the title, as it is excessively lengthy and lacks information about the type of study. Kindly take into consideration my suggestion, or provide an explanation if you choose not to address this comment.

Reviewer 3 Answer-1: Revised. ‘Effects of Swedish massage at different times of the day on dynamic and static balance in Taekwondo Athletes’

Reviewer 3 Comment-2: "Please use the template provided by Healthcare to complete the abstract. Abstract must be divided by sections". This point holds significant importance. It is imperative to adhere to the Guide for Authors of the respective journal when submitting a manuscript. In the case of Healthcare, a template is provided for adherence. However, I observe a deviation from these guidelines in your submission, particularly in the structure of the abstract. I kindly request that you address this issue in your revision. Thank you very much again for allowing me to review your manuscript.

Reviewer 3 Answer-2: We read the Guide for Authors again.  ‘The abstract should be a single paragraph and should follow the style of structured abstracts, but without headings: 1) Background: Place the question addressed in a broad context and highlight the purpose of the study; 2) Methods: Describe briefly the main methods or treatments applied. Include any relevant preregistration numbers, and species and strains of any animals used; 3) Results: Summarize the article's main findings; and 4) Conclusion: Indicate the main conclusions or interpretations. The abstract should be an objective representation of the article: it must not contain results which are not presented and substantiated in the main text and should not exaggerate the main conclusions.’. Thank you for your attentions. Revised.

Reviewer 4 Report

Comments and Suggestions for Authors

Thank you very much for all your revisions. However, there are still small issue need to be revised as followed.

-Please kindly present the similar number of decimal for data L. 116-118.

-Please check the consistency of data presented in the Table, Such as Es Vs ES and the level of F,p,ES for Dynamic Balance Double Foot.

-Please provide the full word of avgfl, avgfr, avgfb, and avgff underneath the Table 2.

-Please also reconsider to mention that only 10-15 min massage significantly improve dynamic balance as the results showed that 5 min also substantially improved. Please explain.

Thank you very much.

Author Response

REVIEWERS’ COMMENTS AND AUTHORS RESPONSE

Paper title: Effects of Swedish massage at different times of the day on dynamic and static balance in Taekwondo Athletes

Dear Editor,

The authors would like to thank the reviewers for their precious time and invaluable comments. We have carefully addressed all the comments. Our detailed, point-by-point responses to the reviewer comments are given below. Additionally, we have carefully revised the manuscript to ensure that the text is optimally phrased and free from typographical and grammatical errors. Best regards,

Author's Reply to the Review Report (Reviewer 4)

Thank you very much for all your revisions. However, there are still small issue need to be revised as followed.

Reviewer 4 Comment-1: Please kindly present the similar number of decimal for data L. 116-118.

Reviewer 4 Answer-1: Thanks for the comment and revised.

Reviewer 4 Comment-2: Please check the consistency of data presented in the Table, Such as Es Vs ES and the level of F, p, ES for Dynamic Balance Double Foot.

Reviewer 4 Answer-2: Revised.

Reviewer 4 Comment-3: Please provide the full word of avgfl, avgfr, avgfb, and avgff underneath the Table 2.

Reviewer 4 Answer-3: Thanks and added.

Reviewer 4 Comment-4: Please also reconsider to mention that only 10-15 min massage significantly improve dynamic balance as the results showed that 5 min also substantially improved. Please explain.

Reviewer 4 Answer-4: Thank you so much for your comment and it is explained in the relevant section. When applied for 10 and 15 minutes, Swedish massage enhances dynamic balance performance more than a 5-minute Swedish massage. Therefore, we discussed the effects of the massage duration in the debate section.